# Toward High-Energy-Density Aqueous Lithium-Ion Batteries Using Silver Nanowires as Current Collectors

**DOI:** 10.3390/molecules27238207

**Published:** 2022-11-25

**Authors:** Jingyi Kong, Yangyang Wang, Ying Wu, Liang Zhang, Min Gong, Xiang Lin, Dongrui Wang

**Affiliations:** School of Chemistry and Biological Engineering, University of Science and Technology Beijing, Beijing 100083, China

**Keywords:** aqueous lithium-ion battery, silver nanowire, current collector, energy density, molecular-crowding electrolyte

## Abstract

The lack of suitable lightweight current collectors is one of the primary obstacles preventing the energy density of aqueous lithium-ion batteries (ALIBs) from becoming competitive. Using silver nanowire (AgNW) films as current collectors and a molecular crowding electrolyte, we herein report the fabrication of ALIBs with relatively good energy densities. In the 2 m LiTFSI–94% PEG–6% H_2_O solution, the AgNW films with a sheet resistance of less than 1.0 ohm/square exhibited an electrochemical stability window as broad as 3.8 V. The LiMn_2_O_4_//Li_4_Ti_5_O_12_ ALIBs using AgNW films as the current collectors demonstrated an initial energy density of 70 Wh/kg weighed by the total mass of the cathode and anode, which retained 89.1% after 50 cycles.

## 1. Introduction

Since the pioneering work of Dahn et al. [1], aqueous lithium-ion batteries (ALIBs) have made tremendous strides in the past two decades as a highly promising electrochemical energy storage technology [2,3,4,5]. However, the use of aqueous electrolytes is a double-edged sword. On the one hand, the high ionic conductivity, non-flammability, and non-toxicity of aqueous electrolytes confer them with intrinsic safety and low costs compared to conventional LIBs employing organic electrolytes. The narrow electrochemical stability window (ESW) of water, on the other hand, severely restricts the working voltages of ALIBs, resulting in low energy densities. Over the past few years, the discovery that aqueous electrolytes containing high concentrations of salts, i.e., water-in-salt electrolytes, can significantly delay the water-splitting reactions has led to intensive research on expanding the ESW, thereby enabling high-voltage ALIBs [6,7,8,9,10]. For instance, ALIBs with energy densities approaching 200 Wh/kg have been proposed, using LiCoO_2_ and sulfur as the cathode and anode, respectively, in conjunction with a water-in-salt electrolyte [10]. Recently, efforts have been devoted to the development of electrolyte systems that enable high-voltage operation with low-salt concentrations, thereby reducing the cost of organic salts. Using polyethylene glycol (PEG) as a molecular crowding agent, a new aqueous electrolyte demonstrated an ESW of up to 3.2 V at a lithium salt concentration as low as 2 m (mol/kg) [11].

Aside from the working voltages and capacities of the active materials, the energy densities of electrochemical batteries are also affected by non-electrode components such as current collectors, separators, binders, and encapsulation. In contrast to the intensive investigations into electrolytes, very little attention has been paid to current collectors for ALIBs. The vast majority of the published works employed a range of metal foils or meshes such as stainless steel, aluminum, titanium, and nickel as current collectors for bearing the active materials. For example, Wang and colleagues examined the ESW of stainless steel in lithium bis(trifluoromethane sulfonimide) (LiTFSI) aqueous solutions, confirming that the ESW could approach 3.0 V as the salt content increased [6]. The Al foils provided an even larger ESW of up to 4.2 V in the 21 m LiTFSI aqueous electrolyte, owing to the slower reactions of hydrogen evolution revolution (HER) and oxygen evolution revolution (OER) caused by the spontaneously formed surface passivation layer [12]. Nonetheless, metal foils/meshes are heavy and susceptible to fatigue fractures when subjected to mechanical deformations [13]. Hu et al. reported a self-standing film constructed of high-temperature welded multi-walled carbon nanotubes (MWCNTs) for ALIB current collectors [14]. The ESW of the film in the 21 m LiTFSI solution was 3.0 V, which is comparable to that of stainless steel. However, the intrinsically low conductivity of carbon-based nanomaterials may diminish the electron transport capabilities of such current collectors [15,16,17]. Identifying ideal current collectors that are lightweight, highly electrically conductive, and chemically inert to both active materials and electrolytes remains an open question for boosting the energy densities of ALIBs.

Herein, we propose the use of silver nanowires (AgNWs) as current collectors for ALIBs. Metal nanowires have been investigated extensively for the fabrication of flexible/stretchable electronic devices due to their unique one-dimensional nanostructure and physical properties [18,19,20]. Among the various metal nanowires, AgNWs have gained the most attention due to their exceptionally high electrical conductivity, good mechanical strength, and modest cost of preparation [21,22,23]. We found that the current collector comprising AgNWs with a sheet resistance of less than 1.0 ohm/sq. provided an ESW of 3.8 V in the 2 m LiTFSI–94% PEG–6% H_2_O electrolyte, which is sufficient for a variety of commercial energy storage materials. The model ALIBs utilizing AgNWs films as the current collectors and LiMn_2_O_4_ (LMO) and Li_4_Ti_5_O_12_ (LTO) as the active materials showed an energy density of 70 Wh/kg when considering the total weight of the cathode and anode. It is hoped that this strategy will assist in the development of optimal current collectors for ALIBs.

## 2. Experimental Section

### 2.1. Preparation of Electrolytes

The molecular crowding electrolyte (2 m LiTFSI–94% PEG–6% H_2_O) was prepared by mixing polyethylene glycol (PEG, with an average molecular weight of 400, Aladdin, Shanghai, China) with ultrapure water (18.2 MΩ) at a weight ratio of 94:6. Then, the mixed solvent was bubbled with argon for 15 min to eliminate dissolved oxygen. Finally, LiTFSI (Sigma-Aldrich, Saint Louis, MO, USA) was dissolved into the PEG–H_2_O mixed solvent at a predetermined concentration of 2 m (mol/kg).

### 2.2. Preparation of AgNW Films

The aqueous dispersion of AgNWs (average diameter and length of 22.5 nm and 24.5 μm, respectively, as shown in Appendix A) was provided by Xingshuo Nano Technology (Suzhou, China). Firstly, the obtained AgNW dispersion was diluted to a concentration of 0.5 mg/mL. Then, certain amounts of the diluted dispersion were vacuum filtrated through PTFE membranes (Millipore, pore size 0.45 µm, diameter 50 mm) and air-dried to form AgNW films with predetermined areal densities.

### 2.3. Fabrication of Electrodes

Li_4_Ti_5_O_12_ (LTO) powders, LiMn_2_O_4_ (LMO) powders, ketjen black (KB) conductive carbon, and acetylene black (AB) were provided by Annuohe New Energy Technology (China). Polyvinylidene fluoride (PVDF) and N-methyl-2-pyrrolidone (NMP) were obtained from Shenzhen Kejing Star Technology (Shenzhen, China). The LMO cathodes, using AgNWs as the current collectors were prepared as follows. The slurry consisting of LMO, KB, and PVDF with a weight ratio of 8:1:1 in NMP was firstly blade-coated onto Al foils (thickness = 16 μm), dried at 80 °C, and roll pressed. Then, the AgNW dispersion was cast onto the top surface of the calendered samples. After drying at 80 °C, the samples were carefully peeled from the Al foil and punched into disc electrodes with a diameter of 16 mm. The areal mass loadings of the LMO and AgNWs were controlled to be ca. 5 mg/cm^2^ and 200 μg/cm^2^, respectively. The LTO anodes on the AgNW films were prepared following a similar approach, except that the slurry recipe was LTO:AB:PVDF = 7:1:2. The mass loading of LTO was about 3 mg/cm^2^. The LMO cathodes and LTO anodes on the Al foils were also prepared as control samples.

### 2.4. Electrochemical Measurements

Cyclic voltammetry (CV), linear sweep voltammetry (LSV), and electrochemical impedance spectroscopy (EIS) were measured on an electrochemical workstation (CHI 660E). The LSV measurements were carried out using a three-electrode configuration, in which the AgNW film, carbon cloth, and Ag/AgCl in a saturated KCl solution were used as the working, counter, and reference electrodes, respectively. The electrolyte was the 2 m LiTFSI–94%PEG–6%H_2_O aqueous solution. CR2032 coin cells were assembled using a glass microfiber membrane (Whatman Ltd., Maidstone, England) as the separator and the 2 m LiTFSI–94%PEG–6%H_2_O as the electrolyte. The cycling tests were performed on a Neware battery testing system at room temperature. For the LMO//LTO battery cycling tests, the full cells were charged and discharged within 1.0–3.5 V. The EIS of the full cells was tested in the frequency range of 10^−2^–10^5^ Hz.

### 2.5. Characterization

The sheet resistance (*R*_s_) of the electrodes was measured using a source meter (Keithley 2450, Beaverton, OR, USA) coupled with a four-probe clamp. The static contact angles were measured on a contact angle tester (JC2000, POWEREACH). Field-emission scanning electron microscopy (SEM) images were recorded on an SU8110 (Hitachi, Tokyo, Japan) at an accelerating voltage of 5 kV. X-ray diffraction (XRD) patterns were measured on a diffractometer (Panalytical Empyrean Series 3, Almelo, Netherlands). The measurements were carried out at 5°/min with Cu Kα radiation (λ = 1.541 Å). The element contents were analyzed by X-ray photoelectron spectroscopy (XPS) on a spectrometer (Thermo Scientific K-Alpha, Waltham, MA, USA).

## 3. Results and Discussion

To examine the electrochemical stability window of the AgNW films in the molecular crowding electrolyte, a series of AgNW films with areal densities in the range of 100–500 μg/cm^2^ were prepared. The AgNWs were densely packed into three-dimensional interconnected conducting networks (see the inset in Figure 1a for an example). As shown in Figure 1a, the sheet resistance gradually reduces from 1.92 to 0.50 ohm/sq. when the areal density of the AgNWs increased from 100 to 500 μg/cm^2^. The AgNW film with an areal density of 200 μg/cm^2^ showed a sheet resistance of 0.97 ohm/sq., which is about half of that with an areal density of 100 μg/cm^2^. The electrochemical stability windows (ESW) of the various conductive materials in the 2 m LiTFSI–94% PEG–6% H_2_O, also referred to as the “molecular crowding electrolyte” [11], are shown in Figure 1b. Using a Au electrode as a reference, the results clearly revealed that this electrolyte was more stable than the well-known water-in-salt (WIS) electrolyte [6]. The ESW of the 2 m LiTFSI–94% PEG–6% H_2_O on the Au electrode achieved 3.3 V. Comparatively, the ESW was only 2.5 V in the 21 m LiTFSI [12]. This improved ESW should be primarily attributed to the inhibition of the hydrogen evolution reaction (HER) of the water by the crowding agent PEG. In this molecular crowding electrolyte, the AgNW films with areal densities of 100 and 200 μg/cm^2^ exhibited significantly wide ESWs of 5.0 and 3.8 V, respectively, which might compete with commonly used Al foils for aqueous LIBs (Appendix A). It is also worth pointing out that the ESW of the AgNW films decreased dramatically when their areal densities surpassed 300 μg/cm^2^. Additionally, the cathodic peaks between 2.0 and 3.0 V vs. Li^+^/Li, which should be ascribed to the reduction of silver oxides on the NW surfaces [24], gradually increased as the AgNWs accumulated. Due to its low sheet resistance and reasonably wide ESW, the AgNW film with an areal density of 200 μg/cm^2^ was selected as the current collector candidate for further investigations. In addition, the ESWs of the AgNW films in some other aqueous electrolytes, such as 2 m LiTFSI and water-in-salt-type electrolytes, are much narrower than those in the molecular crowding electrolyte (as shown in Appendix A).

The primary property of appropriate current collectors is that they remain electrochemically inert within the operating potential window when specific active materials are loaded onto them. We performed CV tests in a three-electrode baker cell to confirm the electrochemical stability of the AgNW films. Figure 2a,b show the CV curves of the selected AgNW film of 200 μg/cm^2^ at 3.7–4.7 V and 1.3–3.0 V vs. Li^+^/Li for 20 cycles. These two potential windows correspond to the working windows of LMO and LTO, two commercially available active materials for aqueous LIBs. Except for the first run, the AgNW film remained relatively stable during the 20 CV scans. The curves were stable when cycled within the range of 3.7–4.7 V vs. Li^+^/Li, with the exception of an apparent anodic peak in the first positive scan, which could be attributed to the oxidation of Ag [25,26]. Furthermore, as the scan progressed, the polarization current gradually decayed. When cycled at relatively low potentials within 1.3–3.0 V vs. Li^+^/Li, the AgNW film exhibited a cathodic peak corresponding to the reduction of silver oxides, which vanished in subsequent scans. The HER reaction, on the other hand, was always present, despite the polarization current remaining relatively low (~50 μA/cm^2^). Figure 2c shows a comparison of the XRD curves of the AgNW films before and after the CV scans. The patterns appeared almost identically for all three samples. Diffraction peaks at 38.1°, 44.3°, 64.5°, 77.6°, and 82.4° were always visible, corresponding, respectively, to the planes (111), (200), (220), (311), and (222) of typical faced face-centered cubic Ag crystals [27,28]. The oxidation of the AgNWs within 3.7–4.7 V vs. Li^+^/Li was extremely harmful because the oxides produced on the surfaces severely reduced the electrical conductivity of the AgNW film. Furthermore, the oxides can dissolve in the aqueous electrolyte, possibly causing the AgNW current collector to degrade or even fail. Therefore, XPS and SEM analyses of the AgNW film were carried out to investigate its electrochemical stability within such a potential window. Appendix A displays the XPS survey scan profiles of the AgNW film before and after the CV tests, which were very similar to each other. Figure 2d,e exhibit high-resolution profiles of Ag 3d and O 1s, which confirm that the atomic contents of Ag and O on the AgNW surfaces remained nearly constant after 20 cycles of CV scans [29,30,31,32]. Furthermore, the SEM image in Figure 2f shows that the morphology of the AgNW percolative networks had not changed. These findings are consistent with the CV results. All of the above results indicate that the AgNW film is electrochemically inert in both potential windows and can be used as a current collector to load LTO and LMO.

To investigate the effect of the AgNW films as current collectors for aqueous LIBs, a typical spinel LMO and LTO were selected as the cathode and anode active materials, respectively [33,34]. The particle sizes, XRD patterns, and morphologies of the two types of powders are depicted in Appendix A. The electrodes were fabricated by coating AgNW dispersion on the LMO//LTO electrodes prepared using the standard slurry-casting technique and then by removing the supporting Al foils. As exhibited in Appendix A, the AgNW films (200 μg/cm^2^) on top of the LMO and LTO electrodes had the same morphology as those produced by vacuum filtration. In addition, it should be mentioned that press rolling prior to AgNW dispersion coating is crucial for fabrication. Appendix A demonstrates that after press rolling, the surfaces of the LMO- and LTO- electrodes became smoother and had improved wettability to the aqueous AgNW dispersion, which ensured the uniform distribution of the AgNWs on their surfaces. The CV curves of the resultant LMO and LTO electrodes in the molecular crowding electrolyte are shown in Figure 3a,d. The CV measurements were carried out in a three-electrode cell at a scan rate of 0.1 mV/s using an oversized activated carbon electrode as the counter and an Ag/AgCl electrode as the reference. The CV curves in Figure 3a display two pairs of redox peaks, which are indicative of the insertion/extraction of Li ions from LMO [35,36]. Note that the area of the CV curve of the AgNW-based electrode is larger than that of an Al foil-based electrode, suggesting that the AgNW-based current collector can induce higher specific capacities of LMO. A similar trend was also found in the CV curves of the LTO electrodes using the two different current collectors [37]. As shown in Figure 3d, the electrode using the AgNWs as the current collector offered a larger CV loop than that using Al foil. Figure 3b,e give the GCD curves for the LMO- and LTO- electrodes in the three-electrode cells. Compared with the electrodes on Al foil, the use of AgNWs as the current collector was found to efficiently enhance the specific capacity and cycling stability of both the LMO and LTO electrodes. Under a current density of 14.8 mA/g, the LMO electrode on AgNWs showed a discharged capacity of 101 mAh/g (based on the mass of the LMO) during the first cycle, which remained with nearly no change over 30 charging/discharging cycles. In contrast, the LMO electrode on Al foil decayed rapidly from 89.5 mAh/g to 34.6 mAh/g during 30 charging/discharging cycles. The LTO electrodes, however, showed relatively stable discharge capacities when cycling at 17.5 mA/g, regardless of whether the current collector was AgNWs or Al foil. The LTO on AgNWs had a specific capacity of 125.1 mAh/g (based on the mass of the LTO), which gradually decreased to 109.5 mAh/g after 30 cycles. When utilizing Al foil, the LTO electrode showed extremely low charging/discharging efficiencies, while the discharged capacities were comparable to those on AgNWs. Figure 3c,f show cross-sectional SEM images of the LMO and LTO electrodes on AgNWs, respectively. It can be seen that the AgNWs were compactly packed on the LMO and LTO electrodes, allowing for strong connections between the active material powders and the current collector.

We also investigated the electrochemical properties of the aqueous full cells using LMO on AgNWs and LTO on AgNWs as the cathode and anode, respectively. Again, the molecular crowding electrolyte was used as the electrolyte. The capacity ratio was controlled at about 3:2 (cathode:anode). As shown in Figure 4a, the CV curve of the resultant full cell exhibited a pair of redox peaks located at around 3.0 V and 2.0 V, respectively. The results indicated that the transport of Li ions between the cathode and the anode supported by AgNWs is highly reversible. Figure 4b shows the galvanostatic charge–discharge curves of the full cell based on the AgNW-based current collectors at a current density of 17.5 mA/g (based on the mass of the LTO). The cell had an initial specific capacity of 39.6 mAh/g (based on the total mass of active materials on the cathode and anode) and showed capacity retention of 89.1% over 50 charge/discharge cycles. In comparison, the full cell using Al foil as the current collectors (results shown in Appendix A) showed a higher initial specific capacity (58.2 mAh/g) but worse cycling stability. The discharge capacity rapidly decayed to 29.1 mAh/g in 50 cycles. Figure 4c and Appendix A demonstrate the EIS spectra of the LMO//LTO full cells using AgNWs and Al foils as the current collectors before and after the galvanostatic cycling. The full cell using the AgNW-based current collector always exhibited a smaller charge transfer resistance (*R*_ct_) than that using the Al foil-based current collector, which can be attributed to the tight contact of the AgNWs with the active material particles and the intrinsically low internal resistance of AgNWs.

Figure 4d compares the energy densities and Coulombic efficiencies of the two LMO//LTO full cells using different current collectors. The energy density was calculated based on the total mass of the cathode and anode, including the mass of the active materials, conductive carbon, polymer binder, and current collectors. The total mass of the two electrodes for the AgNW-based cell was 35.6 mg, which was much lower than that of the Al foil-based cell (53.6 mg). Therefore, the two cells exhibited similar initial energy densities, although the initial specific capacity of the AgNW-based cell was relatively lower. The full cell using AgNWs as the current collectors showed an energy density of 69.3 Wh/kg, which was maintained at 61.7 Wh/kg after 50 cycles. These values are better than most ALIBs using metal foils/grids as current collectors (see Appendix A) and are competitive for current commercial aqueous batteries such as lead-acid batteries [38,39]. In addition, in terms of the Coulombic efficiency, the AgNW-based full cell showed both more stable efficiency values and also a better initial efficiency, as high as 75.1%, which is far greater than that of 55.3% in the case of the Al foil-based full cell. Moreover, the cost of the AgNW film used in this study was approximately 40 USD/m^2^ at an areal density of 200 μg/cm^2^, which is approximately 50 times that of conventional Al foils. Through large-scale wet-chemical synthesis, the price of AgNWs is anticipated to decrease significantly.

## 4. Conclusions

In this work, we investigated the feasibility of using AgNWs as current collectors for aqueous LIBs. In a molecular crowding electrolyte, the AgNW films with an areal density of 200 μg/cm^2^ exhibited an electrochemical stable window of 3.8 V. By selecting LMO and LTO as the representative active materials, the AgNW-based current collectors gave rise to higher specific capacities than conventional Al foils. The resultant LMO//LTO full cells using AgNWs as the current collectors demonstrated an initial energy density of 69.3 Wh/kg (based on the total mass of the cathode and anode) and a Coulombic efficiency of 75.1% with a capacity retention of 89.1% over 50 cycles. These results confirm that the use of lightweight and highly conductive AgNWs is promising for improving the specific energy density of aqueous LIBs, which thus facilitates their commercialization.

## Figures and Tables

**Figure 1 molecules-27-08207-f001:**
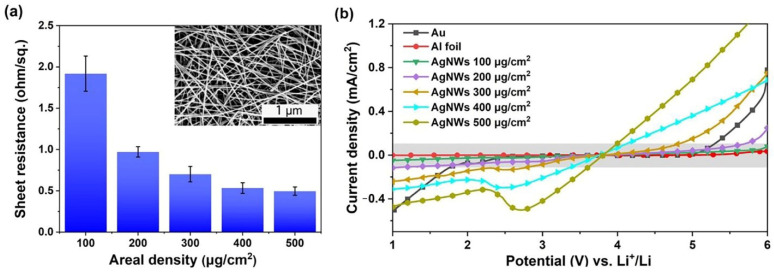
Sheet resistance and electrochemical stability window of AgNW films. (**a**) Sheet resistance of AgNW films with various areal densities. The inset in (**a**) shows one typical SEM image of the AgNW film of 200 μg/cm^2^. (**b**) Linear sweep voltammetry (LSV) curves of various conductive materials under a scan rate of 10 mV/s in 2 m LiTFSI–94%PEG–6%H_2_O electrolyte.

**Figure 2 molecules-27-08207-f002:**
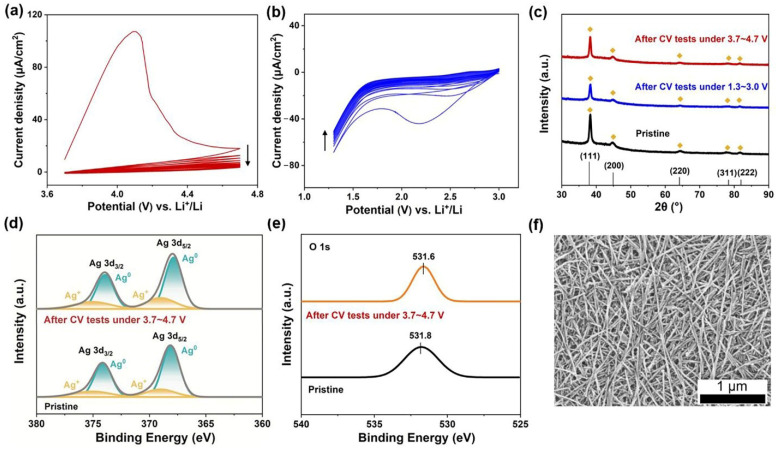
Cyclic stability of AgNW film (areal density of 200 μg/cm^2^) in the molecular crowding electrolyte. (**a**,**b**), Cyclic voltammetry curves of the AgNW film within 3.7–4.7 V vs. Li^+^/Li and 1.3–3.0 V vs. Li^+^/Li at a scan rate of 0.5 mV/s. (**c**) XRD patterns of the AgNW film before and after cyclic voltammetry tests. The standard card of Ag (JCPDS PDF#25-1325) is also presented. (**d**) XPS patterns of Ag 3d of the AgNW film before and after cyclic voltammetry tests. (**e**) XPS patterns of O 1s of the AgNW film before and after cyclic voltammetry tests. (**f**) Typical SEM image of the AgNW film after 20 cycles of cyclic voltammetry tests within 3.7–4.7 V vs. Li^+^/Li.

**Figure 3 molecules-27-08207-f003:**
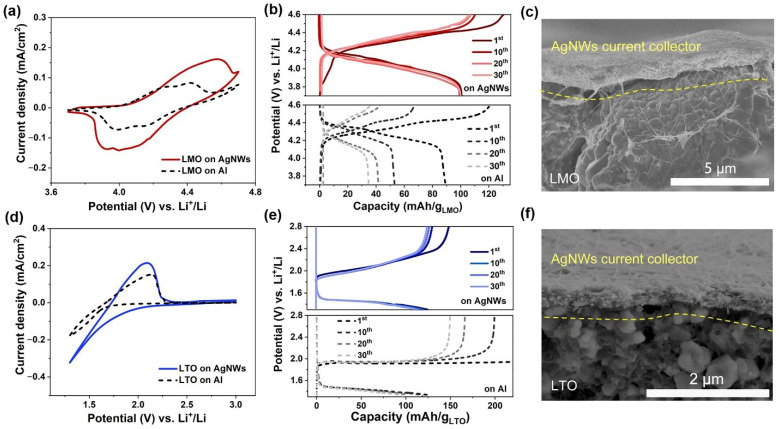
Cyclic voltammetry (CV) curves of (**a**) LMO and (**d**) LTO on AgNWs and Al foil current collectors in 2 m LiTFSI–94% PEG–6% H_2_O at 0.1 mV/s. Galvanostatic charge–discharge (GCD) curves of (**b**) LMO and (**e**) LTO on AgNWs and Al foil current collectors under the current densities of 14.8 mA/g and 17.5 mA/g, respectively. Cross-sectional SEM images of (**c**) LMO on AgNWs and (**f**) LTO on AgNWs.

**Figure 4 molecules-27-08207-f004:**
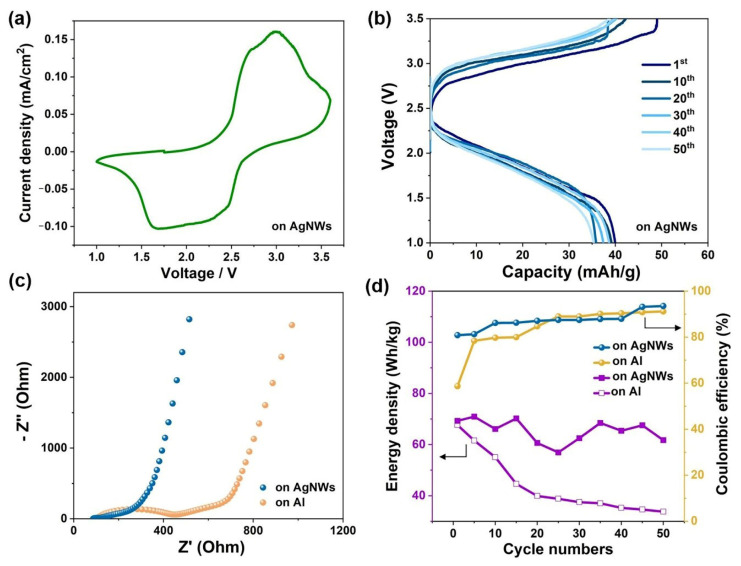
Electrochemical properties of aqueous LMO//LTO full cells using AgNWs as current collectors. (**a**) Cyclic voltammetry curve of the LMO//LTO full cell under a scan rate of 0.1 mV/s. (**b**) Galvanostatic charge–discharge behavior of the LMO//LTO full cell under a current density of 17.5 mA/g (based on the mass of the LTO) for 50 cycles. (**c**) Nyquist plots of as-assembled LMO//LTO full cells using AgNWs and Al foil as current collectors. (**d**) Energy densities (based on the total mass of the cathode and anode) and the Coulombic efficiency of the full cells cycled under a current density of 17.5 mA/g (based on the mass of the LTO).

## Data Availability

Not applicable.

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
