# Peer review of "Toward High-Energy-Density Aqueous Lithium-Ion Batteries Using Silver Nanowires as Current Collectors"

_molecules, 2022, doi:10.3390/molecules27238207_

Round 1

Reviewer 1 Report

This manuscript reported the use of percolative network of silver nanowires as the conductive current collectors for aqueous lithium-ion batteries. This kind of lightweight current collector enables the substantial improvement in energy densities of the entire electrodes including both of the active and non-active components. The strategy is very attractive for boosting the practical applications of aqueous electrochemical batteries. The results shown in the manuscript are clear and supportive for the authors’ claims, and the writing is good. I recommend this manuscript to be published after some minor revisions.

1.      The authors showed that AgNWs are electrochemically stable and give a wide stable window in the special “molecular crowding” electrolyte. Why choose such an electrolyte, and how about the behavior of AgNWs in conventional electrolytes for aqueous lithium-ion batteries or water-in-salt type electrolyte?

2.      Some cost analysis about using this kind of AgNWs as current collectors might be useful to clarify the possibility for large-scale applications in battery industry.

3.      Some typos exist, for instance, the first letter of Coulombic should be capitalized.

Reviewer 2 Report

The LiMn2O4//Li4Ti5O12 ALIBs using AgNW-films as current collectors demonstrate an initial energy density of 70 Wh/kg weighed by the total mass of cathode and anode, which retains 89.1% after 50 cycles.

Do you have compared to the aqueous system in LIBs, such as like CMC/SBR to evaluate the differences ? 

Reviewer 3 Report

The authors proposed using silver nanowires (AgNWs) as current collectors for ALIBs. This work provided an ESW of 3.8 V in 2 m LiTFSI–94% PEG–6% H2O electrolyte, which is sufficient for a variety of commercial energy storage materials. This strategy will speed up research into developing optimal current collectors for ALIBs. The overall level of the article is higher, I suggest accepting this manuscript after minor revisions.

1. Please provide the standard XRD card number in Figure 2c.

2. In Figure 2a, the lines of b are too thick to distinguish the trend of the curve.

3. The English language and style of the manuscript need to be improved.
